# Hydration and Compressive Strength of Activated Blast-Furnace Slag–Steel Slag with Na_2_CO_3_

**DOI:** 10.3390/ma15134375

**Published:** 2022-06-21

**Authors:** Yunfeng Wang, Bo Jiang, Ying Su, Xingyang He, Yingbin Wang, Sangkeun Oh

**Affiliations:** 1School of Civil Engineering, Architecture and Environment, Hubei University of Technology, Wuhan 430068, China; welking@foxmail.com (Y.W.); cbmasuying@163.com (Y.S.); wang_yingbin@163.com (Y.W.); 20151017@hbut.edu.cn (S.O.); 2School of Architecture, Seoul National University of Science and Technology, Seoul 01811, Korea

**Keywords:** blast-furnace slag, compressive strength, rheological behavior, hydration products

## Abstract

Alkali-activated materials (AAMs) are regarded as an alternative cementitious material for Portland cement with regards to sustainable development in construction. The purpose of this work is to investigate the properties of activated blast-furnace slag (BFS)–steel slag (SS) with sodium carbonate (NC), taking into account BFS fineness and Na_2_O equivalent. The hydration was investigated by rheological behavior and pH development. The hydrates were characterized by TG-DTG and XRD, and the microstructure was analyzed by SEM and MIP. Results showed that the rheology of activated BFS-SS pastes was well-fitted with the H-B model and affected by BFS fineness and NC mixture ratio. It was found that BFS fineness and NC ratio played a crucial role in the initial alkalinity of SS-BFS-based pastes. As such, lower BFS fineness and higher NC ratio can dramatically accelerate the formation of reaction products to endow higher mechanical strength of BFS-SS pastes. However, the effect of NC ratio on the microstructure development of BFS-SS based AAMs was more obvious than BFS fineness.

## 1. Introduction

Portland cement contributes in a significant way to the worldwide modernization of cities, but consumes a significant amount of natural resource and energy [1,2]. In recent times, the cement industry and its associations have continuously worked to improve the situation by using wastes to reduce natural resources and energy consumption, greenhouse-gas emissions, and other environmental impacts [3,4,5].

In the past few decades, alkali-activated materials (AAMs) have been regarded as an alternative cementitious material for portland cement from the perspective of sustainable development. Generally, AAMs can be prepared by aluminosilicate precursors such as fly ash, blast-furnace slag, and steel slag [6,7], which not only reduces CO_2_ emissions by 70–80% and saves energy consumption [8], but also gives the materials high mechanical properties, durability, heat resistance, and acid-resistance performance [9,10,11,12].

Blast-furnace slag (BFS), as a by-product of the steelmaking industry, is commonly used as an aluminosilicate precursor of AAMs [13]. The composition of BFS is similar to portland cement, which is mainly CaO, SiO_2_, Al_2_O_3_ with relatively high pozzolanic reactivity [14,15,16]. It is found that the mechanical properties of alkali-activated BFS pastes could be improved by increasing the activator mixture ratio. Although the positive effect on the early-age compressive strength can also be achieved by increasing the curing temperature, it is not beneficial for the continuous development of mechanical properties [17]. The use of weaker alkaline substances such as Na_2_CO_3_ and CaO can significantly increase the amount of hydration products and result in a compact microstructure and high mechanical strength [18]. Due to the excellent performance of alkali-activated BFS, BFS-based products and technologies have gained increasing attention in recent years. The utilization rate of BFS is more than 70% in China [19]. Comparatively, the recycling of other industrial solid wastes has not been researched adequately.

Steel slag (SS) is another by-product of the steel industry. The annual SS output is about 0.1 billion tons in China, but the utilization rate is only about 20% [20,21,22]. Currently, the storage of SS in China has exceeded 1 billion tons, leading to high land occupancy and serious environmental problems. Therefore, it is urgent to carry out the comprehensive utilization of SS. The mineral composition of SS (C_3_S and C_2_S) is similar to that of cement, which endows SS with cementitious properties [23,24,25]. Some literatures have proved that SS is recyclable and valuable especially as the precursor of AAMs [26,27]. Sun et al. found that the alkali-activated SS had a similar hydration process and similar products to cement, but its loose microstructure retards the development of strength, which was only 30–40% of the cement [27]. The combination of SS and BFS can achieve better properties [13,28]. Appropriate substitution of BFS with SS can improve the mechanical properties of hardened pastes, reduce shrinkage, and improve the wear and corrosion resistance of cement and concrete [29]. Additionally, the formation of Ca(OH)_2_ from C_3_S in SS could accelerate the fracture of Si-O-Al and O-Si-O bonds, and the dissolved alumina tetrahedron can react with SO_4_^2−^ and Ca^2+^ to form ettringite, which promote the strength development of SS-BFS binary cementitious materials [30]. The addition of SS also can increase the mechanical properties of alkali-activated fly ash and BFS based fiber-reinforced composite [31].

For AAMs, the types and ratio of alkali activator play a key role on the overall properties [32]. Many researchers used strong bases as the activators, e.g., sodium silicate and sodium hydroxide, to effectively stimulate the activity of pozzolanic materials. Fernandez-Jimenez et al. found that the effect of activators on the mechanical strength of alkali-activated slag mortars was in the order of Na_2_SiO_3_·nH_2_O > NaOH > Na_2_CO_3_ > NaOH [33]. Aliabdo et al. and Tuyan et al. indicated that high strength can be achieved by increasing the molar concentration of sodium hydroxide and sodium silicate [17,34]. However, AAMs activated by strong bases are difficult for industrialization due to low controllability, strong corrosion, high cost, and strict processing requirements [35,36]. Na_2_CO_3_ is more realistic, feasible, and cheaper than the strong bases (e.g., NaOH or sodium silicate), which shows less corrosivity and harm to the environment. In the presence of MgO, Na_2_CO_3_ can effectively activate BFS–fly ash composite, and the compressive strength of mortar can reach 60 MPa at 28 d [4]. Additionally, the combination of CaO and Na_2_CO_3_ can significantly accelerate the formation of hydration products and densify the microstructure of hardened pastes [18]. Ellis et al. think that it is possible to activate BFS with sodium carbonate to meet the requirements of ASTM on initial setting time and compressive strength [37].

## 2. Materials and Experiment

### 2.1. Materials

The main materials used in this work were converter steel slag (SS) with median particle size of 27.1 μm and basicity of 2.94, and granulated blast-furnace slag (BFS, named as Raw) with median particle size of 15.0 μm, both from Baowu Wuhan Iron and Steel Group Co., Ltd. (Wuhan, China) Different particle sizes of BFS, with median particle size of 7.13 μm (V600), 5.62 μm (V800) and 3.03 μm (V1000), were obtained by adjusting the airflow of cyclone dust collector. The chemical composition of SS and BFS is depicted in Table 1. The particle-size distribution and XRD results of SS and BFS are presented in Figure 1 and Figure 2, respectively.

Sodium carbonate (NC) as the activator was purchased from China National Pharmaceutical Group Co., Ltd. (Beijing, China) The river sand with fineness modulus of 3.0 was applied to produce mortars.

### 2.2. Experiments

#### 2.2.1. Mixture Preparation

Six different mixtures were prepared to investigate the effect of NC ratio and BFS fineness on the mechanical and microstructure development of AAMs, as shown in Table 2. Binders were composed of SS and BFS, and SS content was 30% of the binders. The water-to-binder weight ratio and binder-to-sand weight ratio were both set to be 0.5. The sodium-oxide equivalent (marked as NC) was selected as 1%, 3%, and 5% of the binder.

Mortars were prepared as follows: SS, BFS, and Na_2_CO_3_ were premixed for about 1 min; water was added into the above mixture followed by stirring for 2 min at low speed; river sand was slowly added and stirred for another 2 min at high speed; afterwards, the prepared fresh mortar was poured into a steel mold with dimensions of 40 × 40 × 40 cm^3^ and cured at the standard condition (R.H. > 95%, 20 ± 2 °C) until their testing ages.

Moreover, pastes without sand were also prepared for the sake of hydration products and microstructure analysis. Hardened samples were crushed into small pieces and immersed into acetone to prevent further reaction.

#### 2.2.2. Methods

The compressive strength of hardened specimens was tested by using TYE-300F mechanical-properties-testing system with a loading rate of 2.4 kN/s. The average values of six specimens were calculated as the compressive strength.

The rheological property of specimens was measured by BROOKFIELD RST-SST rheometer with FTK-RST sample adapter and CC3-40 spindle. The shear rate was increased from 0 to 200 s^−1^ in 30 s and the fresh sample was presheared for a duration of 120 s at 200 s^−1^. During the determination process, increase the shear rate from 0 to 200 s^−1^ in 90 s and then reduce the shear rate from 200 to 0 s^−1^ in 90 s, as shown in Figure 3.

For pH value of leaching solution, 5 g BFS-SS blended with appropriate quantity of Na_2_CO_3_ was added to 50 mL of deionized water; the mixture was stirred for 5, 15, 30, 60, 120, and 240 min and then filtered by 45 μm filter to obtain leaching solution. The pH value was measured by METTLER pH meter. The pH value of BFS-NC blend without SS was measured to be 11.2.

The crystalline phases of hydration products were determined by Bruker D8 Advance X-ray diffractometer (XRD) with CuKα radiation, taking into account a scanning rate of 2°/min and 2θ range of 5–60°.

Thermal gravimetric analysis of specimens was performed based on a simultaneous thermal analyzer (German, STA449c). Approximately 15 mg powder was weighed for this measurement and heated from 50 °C to 800 °C at a rate of 10 °C/min.

The variation of microstructure for specimens was measured using QUAN-TAFEG 450 Field Emission Scanning Electron Microscope (SEM). Carbon layer was coated to increase the electrical conductivity of specimens.

The pore characteristic of hardened pastes was investigated by using Mercury intrusion porosimetry (POREMASTER 60 mercury injection apparatus). Before the test, specimens were dried at 50 °C for 2 days to ensure the specimen was free from any wetness.

## 3. Results and Discussions

### 3.1. Rheological Properties

The rheological properties of AAMs are crucial for its practical application in construction. Thus, the effect of BFS fineness and NC ratio on the rheological behavior of BFS-SS based AAMs pastes was investigated, and the results are shown in Figure 4. The variation of shear stress versus shear rate was analyzed by mathematical rheological model to determine the rheological behavior of all specimens. In the previous research [38,39], the rheological behavior of NaOH-activated BFS-based AAMs fitted well with the Bingham model, and the specimens with water-glass as an activator fitted well with the Herschel–Bulkley (H-B model). By comparison, the results obtained in this work presented a nonlinear relationship between shear rate and shear stress, and the down ramp fitted H-B (Equation (1)) model with correlative coefficient (R^2^) higher than 0.99.
τ = τ_0_ + kγ^n^(1)
where τ is the shear stress, τ_0_ is the yield stress, k is the consistency coefficient, γ is the shear rate, and n is the fluidity index. The parameters are depicted in Table 3.

It can be seen that the fluidity index of all pastes was smaller than 1, indicating that all pastes belonged to shear-thinning fluid. Yield stress is the minimum shear stress for the pastes of cementitious materials to start flowing, which is produced by the cohesive network formed by particles and the adhesion and friction between particles. The paste with raw BFS exhibited a yield stress of 5.73 Pa, which increased to 8.27 Pa, 13.09 Pa and 14.72 Pa when BFS fineness reduced to 7.13 μm (V600), 5.62 μm (V800) and 3.03 μm, respectively. The reduction of BFS fineness led to more reaction products and flocs in the pastes. Furthermore, the finer the BFS is, the less free the water is. Hence, the friction and adhesion between BFS particles increased. From Figure 4b and Table 3, it is obvious that the yield stress of pastes dramatically increased with NC ratio, for which the rise of alkalinity may cause more depolymerization of BFS to form C-(N)-S-A-H gel.

The variation of shear stress of all pastes (during the first 120 min) at a constant shear rate of 50 s^−1^ is shown in Figure 5. The shear stress of the specimen with raw BFS was in the range of 20–12 Pa, which decreased slightly during the first 90 min, and then increased slightly until the end with a turning point of 12.67 Pa around 90 min (as shown in Figure 5a). With the decrease in the BFS particle size, the shear stress increased significantly and the turning point appeared much earlier. Furthermore, the pastes with fine BFS presented higher shear stress variation. As for V10-3%, the shear stress increased rapidly after the turning point around 30 min, which apparently differed from Raw-3% and indicated the high hydration rate. The differences are associated with the fineness of BFS, which is the key factor for the depolymerization of BFS and the reaction activity between silicate and calcium ions from SS and BFS.

Figure 5b shows the effect of NC ratio on the shear stress of BFS-SS based pastes. It can be seen that the shear stress and the degree of variation increased as the NC ratio increased. Moreover, the turning point appeared earlier when a higher activator ratio was applied. This could be due to the quick reaction of NC and Ca ions derived from SS/BFS. Thus, more silicate and aluminate network structures, mainly as [SiO_4_] and [AlO_4_], were formed at earlier ages. Essentially speaking, the variation of turning point versus NC ratio was caused by the change in pH value in the AAMs, which will be analyzed later.

### 3.2. pH Analysis

The effect of BFS fineness and NC ratio on pH value of the leaching solution of BFS-SS based AAMs pastes was measured to evaluate the alkalinity of the pore solution to the activation of BFS. The evolution of pH value as a function of time is listed in Figure 6. It can be seen that pH of Raw-3% (with raw BFS) was 12.28 at 5 min and increased with time. Comparatively, pH of pure BFS was 11.2, indicating the crucial role of SS on the initial alkalinity of SS-BFS based AAMs. For specimens containing V600, V800, and V1000, the pH value increased significantly with the reduction in WGP fineness, due to the fact that the smaller BFS particle led to more dissolved calcium ions. Moreover, the alkalinity increased over time firstly and then gradually went down after 60 min. In the increasing stage, the alkali metal ions leached from SS and BFS reacted with NC to improve the pH value of the paste solution, as shown in Equation (2). However, in the decline stage, C-N-S-A-H formed and consumed a large amount of alkali metal ions and OH-, leading to the decline of pH as shown in Equation (3). Furthermore, the pH value of specimens with V1000 decreased more obviously than that of specimens with V800, implying that the reactivity of V1000 is higher than that of V800. A similar result was also found in the investigation of pH value of BFS-SS pastes with different NC ratio. Specifically, the pH value of all specimens increased firstly and then decreased. The aforementioned phenomenon was in accordance with the rheological results.
Ca^2+^ + Na_2_CO_3_→CaCO_3_ + Na^+^(2)
Ca^2+^ + SiO_3_^2−^ + AlO^2−^ + Na^+^ + OH^−^ + H_2_O→C-N-S-A-H(3)

### 3.3. Compressive Strength

The compressive strength of mortars as a function of BFS fineness and NC ratio is presented in Figure 7. As can be seen from the results, the compressive strength of specimens increased with the decrease in BFS particle size. After 3 days reaction, the compressive strength of specimens with V600, V800, and V1000 BFS increased by 15.9%, 24.2%, and 50.0% respectively, compared with that of specimens with ordinary BFS. There is a common understanding that the finer BFS could remarkably improve the mechanical property of BFS-based AAMs, especially for its early strength. In our previous work [40], the compressive strength of cement with ultrahigh-volume ultrafine BFS at 3 d increased by 31.4–120.0% when the particle size reduced from 18.2 μm to 2.98 μm. The increasing rate of BFS-SS strength was obviously lower than that of BFS-cement results. Interestingly, the 3 d and 7 d compressive strength of samples containing BFS-SS is obviously higher than that of AAMs with only BFS [41]. This may be attributed to the increase of alkalinity caused by the dissolving of SS.

The compressive strength of the specimens increased progressively with curing age. At 28 d, the compressive strength of specimens with V600, V800, and V1000 BFS-SS was 31 Mpa, 32.6 Mpa, and 38.7 Mpa, respectively. The result of NC-activated BFS mortars with V600, V800, and V1000 BFS was 29.7 Mpa, 41 Mpa, and 46.3 Mpa, respectively, as described in our previous work [42]. It can be seen that most of the results of BFS-SS specimens were lower than that of BFS specimens, which may be associated with the low pozzolanic activity of SS. The above results indicated that the improvement of the mechanical properties of BFS-SS-based AAMs in the early period (<7 d) was greater than that of pure BFS. However, in the later period (7–28 d), SS showed less facilitation on strength development than BFS.

From Figure 7b, the mechanical property was significantly improved with the increase in NC ratio. It was seen that the specimens with 1% NC showed poor mechanical strength during the investigated curing ages. Nevertheless, a high early compressive strength could be achieved after 3% NC was incorporated, and the maximum value at 28 d was 39.8 Mpa when 5% NC was used.

From the point of view of fineness, BFS with lower particle size provided more reaction sites to form more hydration products, resulting in the compact structure of hardened pastes. In terms of NC, the increase in NC content improved the alkalinity dramatically and activated the pozzolanic reaction remarkably.

### 3.4. XRD Analysis

The mineralogical phases of BFS-SS-based AAMs were investigated by XRD, as shown in Figure 8. It can be seen that C-(N)-S-A-H as one of the main reaction products of BFS-SS AAMs can be detected in all curves. Furthermore, newly formed products, e.g., hydrotalcite and calcite, were also observed. Calcite was derived from the reaction between activator (NC) and calcium ions in SS/BFS. Hydrotalcite contained a high content of magnesium and was often detected as one of the main hydration products of BFS-based AAMs [43].

It should be noted that the fineness of BFS and ratio of NC did not cause changes in the types of hydration products. However, the peak intensity of C-(N)-S-A-H increased with the decrease in BFS fineness, which was in accordance with the mechanical analysis. The small BFS particles have high pozzolanic reactivity and are in favor of the precipitation of C-(N)-S-A-H. Moreover, the formation of C-(N)-S-A-H was obviously improved with the increase of NC ratio. Pore solution with higher alkalinity accelerated the depolymerization of BFS-SS to form C-(N)-S-A-H. The reduction of C_3_S and C_2_S peak intensity was also detected, implying that reducing the particle size of BFS or increasing NC content can stimulate the reaction of SS. The crystalline peak of hydrotalcite declined in the order of Raw > V600 > V800 > V1000 due to the reduction in magnesium content in BFS. In contrast, the intensity of the hydrotalcite peak increased with increasing NC ratio, proving that hydration degree was improved with rising alkalinity.

### 3.5. TG-DTG Analysis

TG results of SS-BFS-based AAMs after 3 d and 28 d reaction are shown in Figure 9. Evidently, all curves included three weight-loss stages: 50–250 °C, associated with the dehydration of C-(N)-S-A-H and the evaporation of free water [44]; 300–400 °C, representing the decomposition of hydrotalcite [45]; 500–700 °C, related to the decomposition of calcite [41]. The weight losses at different temperatures were calculated to evaluate the effect of BFS fineness and NC ratio on the hydration of SS-BFS blends (as shown in Table 4). It can be seen that the weight loss of each stage increased with the fineness of BFS, and the sample with V1000 BFS presented the highest value. Additionally, the increasing of the NC ratio also significantly enhanced the mass loss of each stage. The above results indicated that the hydration process was accelerated as the particle size decreased or the alkalinity increased even after 28 d reaction. The chemically bound water (*W*, 50–550 °C) was used to estimate the hydration degree of BFS-based AAMs since it marked the main hydration products, such as C-(N)-S-A-H and hydrotalcite. *W* was calculated as Equation (4), and the results are depicted in Table 4.
(4)$$W=\frac{m_{50}-m_{550}}{m_{550}}$$
where *m*_50_ is the weight of sample at 50 °C and *m*_550_ is the weight of sample at 550 °C.

It was noticed that *W* enhanced with the increment of NC ratio or the reduction in BFS particle size. For instance, at 3 d, the value of *W* for specimen with Raw BFS was 11.08%, which increased to 15.09% when the particle size reduced to 3.03 μm. Similarly, *W* increased from 8.13% to 15.52% as the ratio of NC increased from 1% to 5%. Furthermore, the value of *W* increased with the prolonging of curing time, proving the continuous reaction of SS and BFS.

### 3.6. SEM Analysis

The effect of BFS fineness and NC ratio on the microstructure of BFS-SS based AAMs was studied, and the representative specimens of Raw-3%, V10-3%, V8-1% and V8-5% were imaged (Figure 10). It was obvious that the specimen with raw BFS presented loose and inhomogeneous microstructure, and a large amount of portlandite crystals embedded in the gel products, implying the low reaction degree of this mix. In comparison, the specimen with V1000 BFS presented to be denser and have a more homogeneous microstructure, where a large amount of gel products and only a few unreacted particles with smooth surface were detected. It is proved that the finer BFS particles can promote the formation of gel products to provide more compact structure and higher mechanical strength, which was confirmed in Section 3.3. In terms of NC ratio, the compactness of hardened pastes was increased with the increment of NC ratio. For example, unreacted BFS and loose microstructure can be observed in the specimen with low NC ratio, whereas specimens with high content of activator exhibited more compact gel products and fewer voids.

### 3.7. MIP Analysis

In order to investigate the microstructure of hardened SS-BFS based AAMs, MIP analysis was also conducted on all specimens after 28 d reaction. The results of pore-size distribution are shown in Figure 11. As seen from Figure 11a, the most aperture pore (MAP) was increased with the increasing of BFS fineness. For example, the MAP of specimens with Raw BFS (15.0 μm) was 24.67 nm, which reduced to 16.07 nm, 14.59 nm, and 11.45 nm when the median grain size of BFS decreased to 7.13 μm, 5.62 μm, and 3.03 μm. This is due to the fact that BFS with finer particle size could form more reaction products to refine the pore radius and have better filling effect to compact hardened pastes. Furthermore, the growth of NC ratio also led to the reduction in MAP, indicating that increasing NC ratio was beneficial for compacting the microstructure.

In general [46], the pores of hardened cementitious materials can be divided into four groups according to their radius: gel pores, fine capillary pores, middle capillary pores, and coarse capillary pores. Cumulated pore volume of hardened specimens as the function of BFS fineness and NC ratio is shown in Figure 12. It can be seen that most of the pores were smaller than 50 nm and the curves shifted to the left with the decrease in BFS fineness or the increase of NC ratio, indicating the denser microstructure of the AAMs with finer BFS and higher NC content. The pore-volume distribution was calculated to further quantify the effects of BFS fineness and NC ratio as shown in Figure 13. The volume of gel pores for specimens with fine BFS was obviously larger than that of specimens with coarse BFS. The finer BFS was in favor of the formation of C-(N)-A-S-H gel, which filled the capillary pores and converted them into gel pores with a smaller pore size. Moreover, the volume of gel pores was significantly improved with the increase in NC ratio since more gel products were produced at high alkalinity.

Based on the above analysis, the pore structure can be refined by increasing NC content or decreasing BFS fineness. The effect of NC ratio, however, was more apparent than that of BFS fineness.

## 4. Conclusions

In this study, the influence of blast-furnace slag (BFS) fineness and sodium carbonate (NC) ratio on the properties of steel slag (SS)-BFS-based alkali-activated materials (AAMs) was investigated, and the following conclusions can be drawn:The rheological behavior of NC-activated SS-BFS pastes fitted well with the Herschel–Bulkley (H-B) model. With the decrease in the BFS particle size and the increase of the NC ratio, the shear stress increased significantly and the turning point appeared much earlier.Both BFS fineness and NC ratio can influence the pH level of pore solution. SS played a crucial role on the initial alkalinity of SS-BFS-based AAMs.The hydration process of SS-BFS-based AAMs was greatly accelerated to form more reaction products and generate a higher mechanical property by decreasing the BFS particle size and increasing NC ratio.The pore structure can be refined by increasing NC content or decreasing BFS fineness. Although both approaches can greatly promote the transformation of capillary pores to gel pores, the effect of NC ratio was more apparent than that of BFS fineness.

## Figures and Tables

**Figure 1 materials-15-04375-f001:**
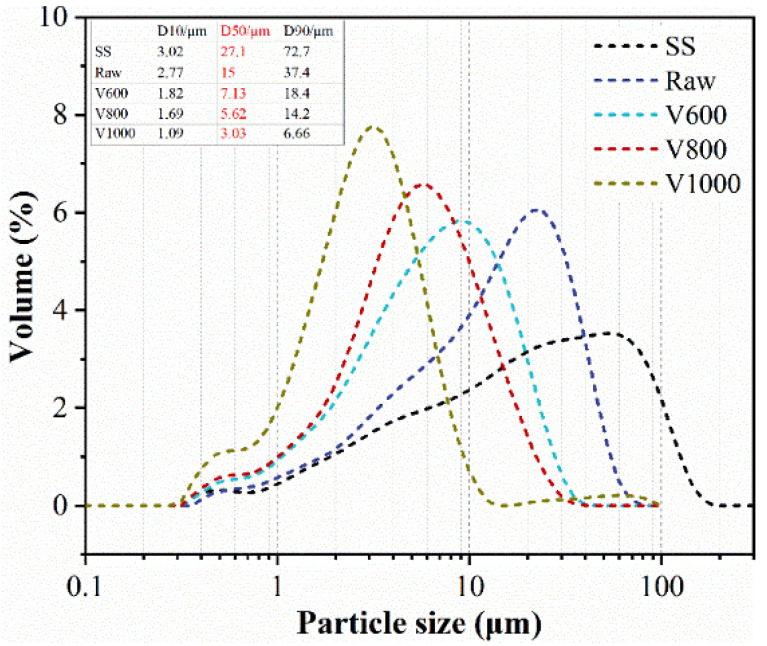
The particle-size distribution of BFS and SS.

**Figure 2 materials-15-04375-f002:**
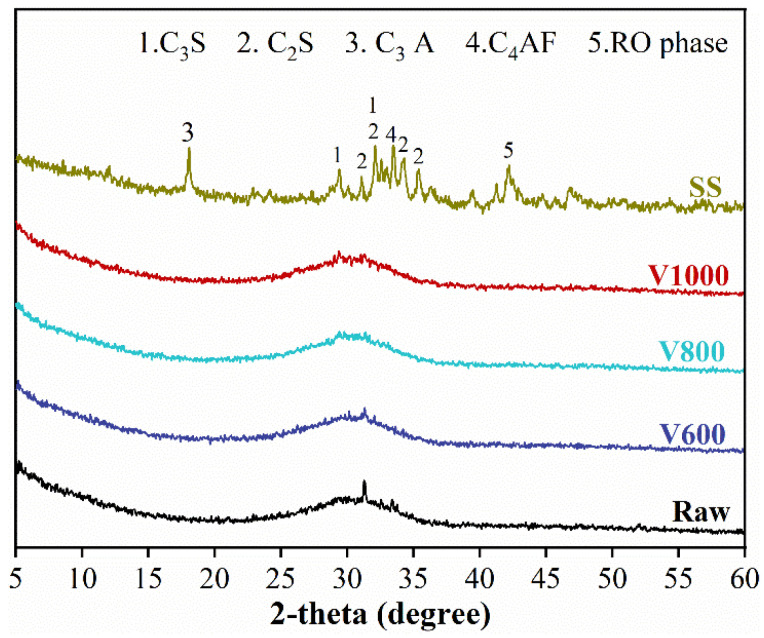
The XRD patterns of BFS and SS.

**Figure 3 materials-15-04375-f003:**
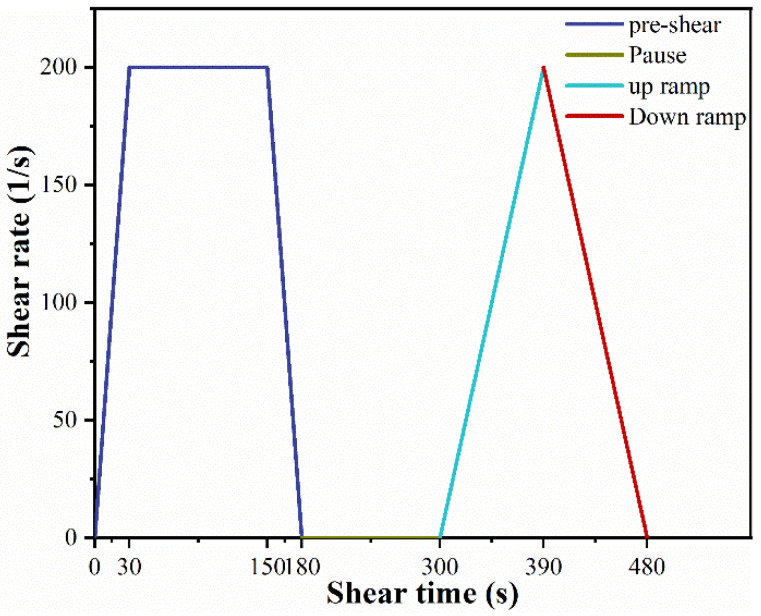
Variation of shear rate with time.

**Figure 4 materials-15-04375-f004:**
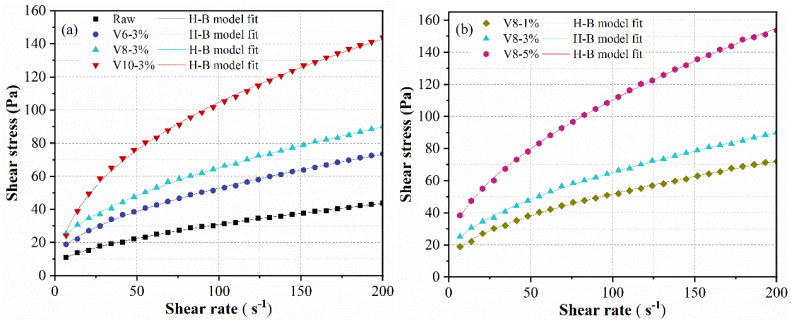
Shear stress (down-ramp) of BFS-SS-based AAMs: effect of BFS fineness (**a**) and effect of NC dosage (**b**).

**Figure 5 materials-15-04375-f005:**
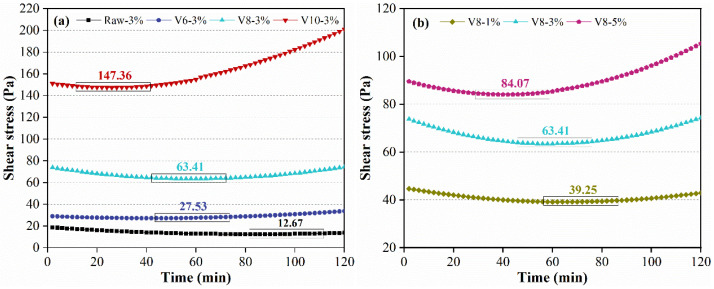
Shear stress of BFS-SS-based pastes at a constant shear rate of 50 s^−1^: effect of BFS fineness (**a**) and effect of NC dosage (**b**).

**Figure 6 materials-15-04375-f006:**
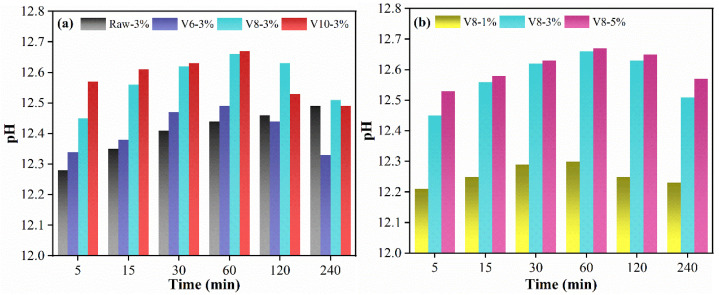
pH value of the leaching solution of SS-BFS-based AAMs: effect of BFS fineness (**a**) and effect of NC dosage (**b**).

**Figure 7 materials-15-04375-f007:**
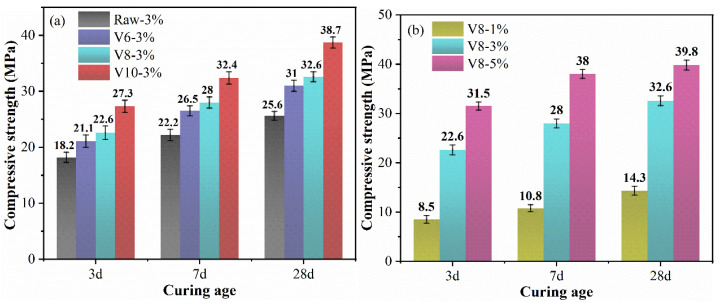
Compressive strength of hardened specimens: effect of BFS fineness (**a**) and effect of NC dosage (**b**).

**Figure 8 materials-15-04375-f008:**
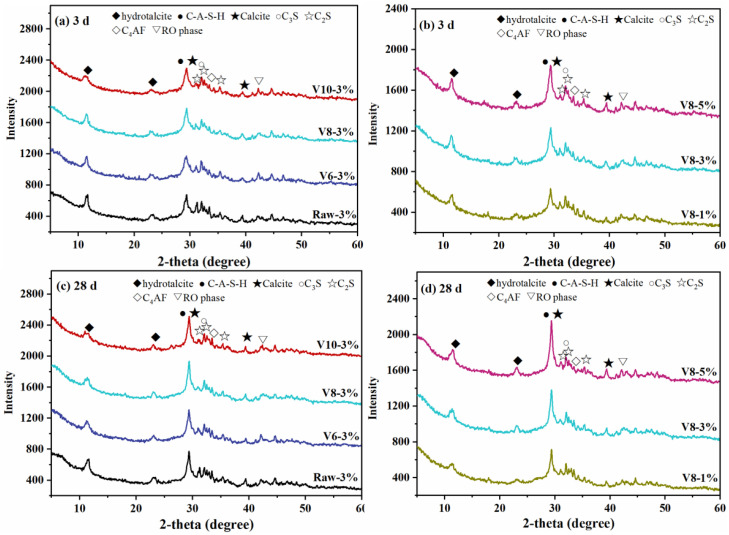
XRD of hardened specimens at 3 d and 28 d.

**Figure 9 materials-15-04375-f009:**
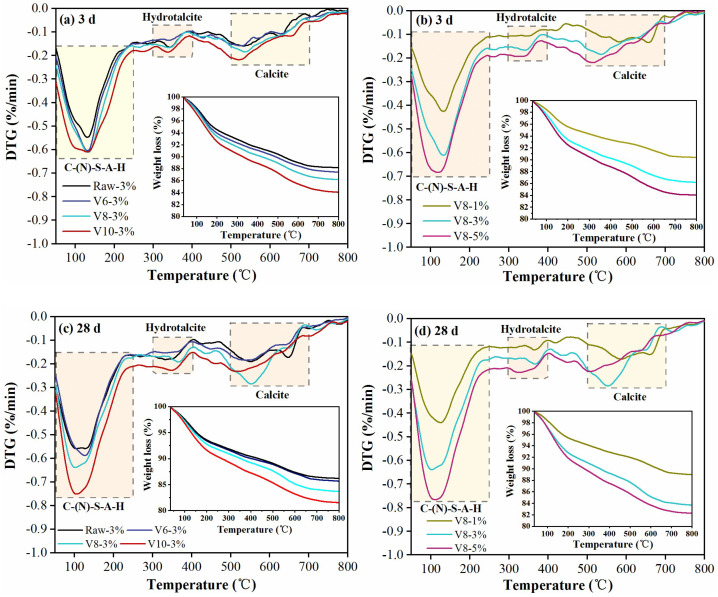
TG−DTG results of hardened specimens at 3 d and 28 d.

**Figure 10 materials-15-04375-f010:**
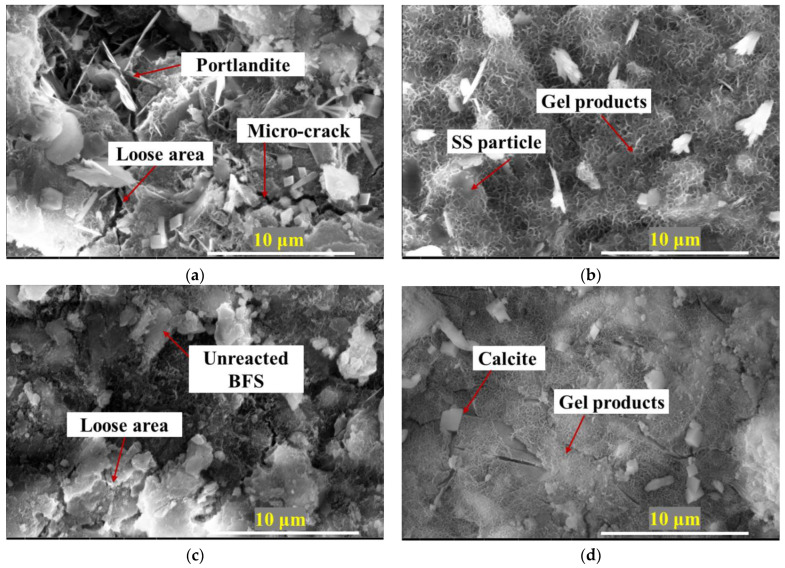
SEM micrographs (**a**) Raw-3%, (**b**) V10-3%, (**c**) V8-1%, (**d**) V8-5%.

**Figure 11 materials-15-04375-f011:**
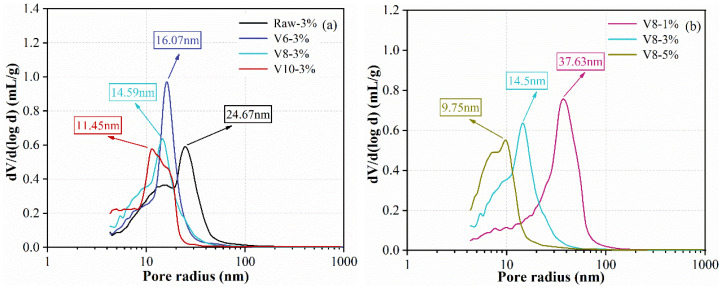
The pore-size distribution of hardened specimens at 28 d: effect of BFS fineness (**a**) and effect of NC dosage (**b**).

**Figure 12 materials-15-04375-f012:**
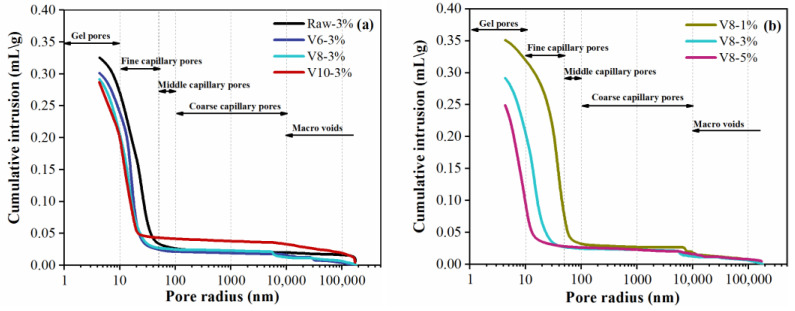
Cumulated pore-size distribution of hardened specimens at 28 d: effect of BFS fineness (**a**) and effect of NC dosage (**b**).

**Figure 13 materials-15-04375-f013:**
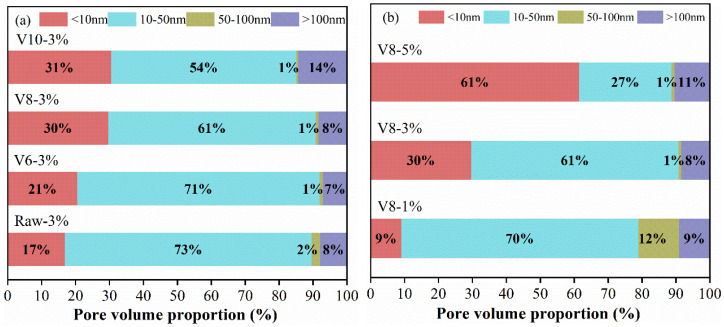
The pore-volume distribution of hardened specimens at 28 d: effect of BFS fineness (**a**) and effect of NC dosage (**b**).

**Table 1 materials-15-04375-t001:** Chemical composition of SS and BFS.

Raw Material	Chemical Composition (%)							
	SiO_2_	CaO	Al_2_O_3_	MgO	SO_3_	K_2_O	Fe_2_O_3_	LOI
SS	14.2	45.5	2.7	4.2	0.4	0.1	21.1	8.0
Raw	30.3	39.2	15.4	9.3	2.4	0.4	0.3	0.8
V600	32.6	40.7	14.8	7.3	2.3	0.5	0.4	0.5
V800	31.8	38.7	15.2	7.3	2.2	0.4	0.3	2.1
V1000	32.7	41.1	16.1	5.9	2.4	0.2	0.3	1.1

**Table 2 materials-15-04375-t002:** The mixture proportions of specimens (wt %).

No	SS	Raw	V600	V800	V1000	NC	b/s	w/b
Raw-3%	30	70	-	-	-	3%	1:2	0.5
V6-3%	30	-	70	-	-	3%		0.5
V8-1%	30	-	-	70	-	1%		0.5
V8-3%	30	-	-	70	-	3%		0.5
V8-5%	30	-	-	70	-	5%		0.5
V10-3%	30	-	-	-	70	3%		0.5

**Table 3 materials-15-04375-t003:** Rheological parameters of the fresh pastes by H-B model.

Sample	Fitted Equation	τ_0_ (Pa)	k (Pa·s^n^)	n	R^2^
Raw-3%	τ = 5.73 + 1.66γ^0.59^	5.73	1.66	0.59	0.999
V6-3%	τ = 8.27 + 3.59γ^0.55^	8.27	3.59	0.55	0.999
V8-3%	τ = 13.09 + 3.92γ^0.56^	13.09	3.92	0.56	0.999
V10-3%	τ = 14.72 + 18.92γ^0.4^	14.72	18.92	0.4	0.999
V8-1%	τ = 9.07 + 3.25γ^0.56^	9.07	3.25	0.56	0.998
V8-5%	τ = 16.5 + 6.99γ^0.56^	16.50	6.99	0.56	0.999

**Table 4 materials-15-04375-t004:** The weight loss of specimens at different temperature ranges.

Age	Temperature Range	Raw-3%	V6-3%	V8-3%	V10-3%	V8-1%	V8-3%	V8-5%
3 d	Δ*m*_1_ 50–250 °C	5.93	6.50	6.99	7.89	4.75	6.99	7.97
	Δ*m*_2_ 300–400 °C	1.35	1.23	1.41	1.58	0.99	1.41	1.68
	Δ*m*_3_ 500–700 °C	2.11	2.17	2.49	2.89	2.11	2.49	2.88
	Δ*m_t_* 40–800 °C	11.66	12.44	13.65	15.71	9.51	13.65	15.75
	*W*	11.08	11.76	12.92	15.09	8.13	12.92	15.52
28 d	Δ*m*_1_ 50–250 °C	6.86	6.76	7.60	8.90	5.00	7.60	8.81
	Δ*m*_2_ 300–400 °C	1.61	1.44	1.72	2.08	1.19	1.72	2.07
	Δ*m*_3_500–700 °C	2.91	2.65	3.59	3.39	2.70	3.59	3.05
	Δ*m_t_* 40–800 °C	13.67	13.23	15.68	17.75	10.70	15.68	17.14
	*W*	12.86	12.95	15.02	17.55	9.09	15.02	17.90

## Data Availability

Not applicable.

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
