# Peer review of "Hydration and Compressive Strength of Activated Blast-Furnace Slag–Steel Slag with Na2CO3"

_materials, 2022, doi:10.3390/ma15134375_

Round 1

Reviewer 1 Report

The authors in the manuscript entitled “Hydration and compressive strength of activated blast furnace slag - steel slag with Na2CO3” investigated the effects of industrial waste on the geopolymer mortar.

The manuscript needs a proofreading by a native proofreader.

Introduction needs a revision as it does not reflect the aim of this work and how it adds on top of the current knowledge. Since there are many studies in this field, what is the novelty of this paper the authors need to elaborate on that.

Figure 1 should be replaced with a high-resolution figure.

Fig 2 should be indexed

Methodology used for SEM and XRD should be noted

Figures to be replaced with a high resolution ones.

Reviewer 2 Report

please see comments embedded in attached file 
